# Population Growth of Fall Armyworm, *Spodoptera frugiperda* Fed on Cereal and Pulse Host Plants Cultivated in Yunnan Province, China

**DOI:** 10.3390/plants12040950

**Published:** 2023-02-20

**Authors:** Kifle Gebreegziabiher Gebretsadik, Ying Liu, Yanqiong Yin, Xueqing Zhao, Xiangyong Li, Fushou Chen, Yong Zhang, Julian Chen, Aidong Chen

**Affiliations:** 1Agricultural Environment and Resources Institute (AERI), Yunnan Academy of Agricultural Science (YAAS), Kunming 650205, China; 2Tigray Agricultural Research Institute (TARI), Mekelle 5637, Ethiopia; 3Chinese Academy of Agricultural Sciences, Beijing 100193, China

**Keywords:** *Spodoptera frugiperda*, host plants, life table, survival rate, maize, wheat, barley, faba beans, soya beans

## Abstract

The fall armyworm, *Spodoptera frugiperda* is a major agricultural pest in China, and has migrated from its continuous breeding area to other parts of China. In our study, the biological behaviors of *S. frugiperda* fed on maize, wheat, barley, faba beans, and soya beans were evaluated in a growth chamber. Results indicated that maize-fed *S. frugiperda* larvae performed well, as evidenced by shorter larva-adult periods, adult pre-oviposition period (APOP), total pre-oviposition period (TPOP), and generation time (*T*), and a higher survival rate, intrinsic (*r*) and finite (λ) rate of increase, and net reproductive rate (*R_o_*), However, *S. frugiperda* larvae performed weakly when fed barley and faba bean plants, as indicated by lower survival rates, *r,* and λ, and longer pre-adult period, TPOP, and *T*. A heavier pupal weight of both sexes was recorded on faba beans (0.202 g) and a lighter weight on barley (0.169 g). Fecundity was higher when fed faba beans and maize, and lower when fed wheat and barley. Thus, maize was the most optimal and barley was the least optimal host plant, followed by faba beans, for *S. frugiperda* larvae growth and development. This study enhances our knowledge of *S. frugiperda* in these host plants and can help in the design of management approaches.

## 1. Introduction

*S. frugiperda* is one of the most common polyphagous insect pests native to America’s tropical and subtropical climates [1,2,3]. It always reproduces in the subtropical and tropical areas of the Americas, but it also migrates to temperate North America during the summer [1,4]. *S. frugiperda* is ranked in the top 10 from the 1187 distractive and invasive arthropod species [5,6]. The pest ranked among the top 10 from the 1187 distractive an invasive arthropod species [5,6]. Due to fluctuations in dietary niches, *S. frugiperda* is erratic and can migrate long distances, with moths capable of traveling more than 100 km in a single night [7,8]. The dissemination and invasiveness of the pest are linked to its unique biological traits, such as lack of diapause, short generation period, high fecundity, high polyphagy, capacity for long-distance migration, and resistance to pesticides, viruses, and Bt toxins [6,7,9,10].

More than 353 species in 76 families, including many economically significant crops, such as maize, sorghum, rice, wheat, barley, oat, millet, ryegrass, soya beans, faba beans, tobacco, tomato, potato, peanut, cotton, sugarcane, alfalfa, and onion, are consumed by *S. frugiperda* [11,12,13]. Weeds such as bent grass, Agrostis ssp.; crabgrass, *Digitaria* spp.; Johnsongrass, *Sorghum halepense*; morning glory, *Ipomoea* spp.; nutsedge, *Cyperus* spp.; pigweed, *Amaranthus* spp.; and sandspur, *Cenchrus tribuloides* are known host plants of *S. frugiperda* in Africa [14]. The invasive *S. frugiperda* has become one of China’s most significant agricultural pests since it was introduced on 11 December 2018 and subsequently bred in tropical and south subtropical regions of China, including Hainan, Taiwan, and the southern regions of Fujian, Guangdong, Guangxi, Guizhou, and Yunnan [4]. The pest was reported in 1524 districts across 26 provinces, with a damaged area of 1125.33 thousand hectares [15]. Currently, about 59.33% of the total area affected by *S. frugiperda* in China is found in Yunnan province [15,16]. Numerous commercial crops have been damaged by this insect, including maize, sorghum, and rice [17], as well as wheat and barley [18]. Additionally, in Yunnan, Hainan, Hunan, Hubei, and other regions of China [16], it has been observed on gramineous weeds such as *Digitaria sanguinalis* (L.) S, *Sorghum sudanense* (Piper), and *Eleusine indica* (L.).

As *S. frugiperda* has been recently introduced to China, it is essential to understand its fundamental biological and ecological characteristics to develop effective control strategies. The biological characteristics of *S. frugiperda* reared on other host and food plants have not yet been studied, except for maize and a few other crops. The effect of *S. frugiperda* damage on crops such as wheat, barley, faba beans, and soya beans in China is less studied. These crops are previously reported host plants of *S. frugiperda* [11] and are commonly grown in many provinces of China, including Yunnan Province. Additionally, because of the subtropical climate of Yunnan Province, these crops are typically planted year-round, which could be sufficient to support the survival and spread of *S. frugiperda*. Differences in host plants significantly impact the population growth, development, and survival rates of phytophagous insect pests [19,20,21,22]. The population growth of any insect pest can be affected by the nutrition and characteristics of its host plant [21,23,24]. The life history features of phytophagous insect pests, such as survival rate, development, reproduction, and population growth are influenced by nutritional variations among host plants on which insects feed throughout the larval stage [20,25,26,27]. For instance, *S. frugiperda* has a generation time of 29.21 d on maize, 42.96 d on tomatoes [17], and 39.04 d on faba beans [20] at 25 1 °C, 60–70% RH, and a 16/8 h (light/dark, L/D) photoperiod. A life table study is crucial to understand population dynamics and pest stations. According to some studies, maize is the ideal host plant for *S. frugiperda*, while other crops such as wheat and sugar cane can also be used as hosts in the absence of maize vegetation in China. Currently, *S. frugiperda* continues to spread and inflict economic damage throughout China, making it imperative to study more about how this pest behaves biologically in the absence of other economically significant crops such as wheat, barley, faba beans, and soya beans. Therefore, it is essential to understand how various host plants affect *S. frugiperda* survival, growth, development, fertility, and population increase to design an all-encompassing management strategy and forecast future population levels. In this study, the survival rate, larval development, and reproductive variables of *S. frugiperda* were assessed after they were fed on maize, wheat, barley, faba beans, and soya beans in a growth chamber, and its demographic features were assessed using the age-stage, two-sex life table. The resulting population life tables will contribute to the development of sound integrated pest management strategies against *S. frugiperda* in maize, wheat, barley, faba bean, and soya bean growing areas.

## 2. Results

### 2.1. Development Time and Adult Longevity of S. frugiperda

As indicated in Table 1, there was no significant difference in *S. frugiperda* egg hatching duration among the five host plants. However, the first instar larval duration of *S. frugiperda* fed on faba beans was significantly longer compared to the other host plants (*p* < 0.001). The second instar larval duration was significantly longer when fed on faba beans, barley, and wheat than when fed on maize and soya beans, and the third instar larval duration on faba beans was significantly longer than on maize, soya bean, and wheat (*p* < 0.001). The fourth, fifth, and sixth larval instars and total larval duration of *S. frugiperda* fed on faba beans and barley were significantly longer compared with larvae fed on the other host plants (*p* < 0.001). Moreover, the pre-pupal duration of *S. frugiperda* was significantly longer on barley (2.27 d) and shorter on maize (1.40 d) (*p* < 0.001). Thus, *S. frugiperda* larvae performed well when fed on maize and performed poorly on faba beans and barley.

The duration in the pupal stage was significantly longer for faba beans (10.75 d), and shorter for maize (9.69 d) and barley (9.95 d) (*p* < 0.001). The pre-adult stage was significantly longer when *S. frugiperda* was fed on faba beans (34.79 d) and barley (33.05 d) and shorter when fed on maize (27.83 d) (*p* < 0.001). The longest adult longevity of both sexes was shown on faba beans (14.59 d) and shortest on barley (10.59 d) (*p* < 0.001). The longevity of female adults was longer than males when fed on maize, wheat, and barley, and shorter when fed on faba beans and soya beans. The total longevity of *S. frugiperda* fed on faba beans (49.38 d) was significantly longer compared to other treatments, but shorter than on maize (39.06 d) (*p* < 0.001). The life span of *S. frugiperda* fed on faba beans was longer compared to the host plants.

The survival rates of *S. frugiperda* at each developmental stage on the five hosts plants are indicated in Table 2. Variations in the survival rates at each larval instar, prepupal, pupal and adult stages were shown among the five host plants. The survival rate was on higher maize and lower on barley at each developmental stage.

### 2.2. Pupae Weight

Male and female pupae showed significant variations in weight when *S. frugiperda* fed on the five host plants (Figure 1). Male pupae fed faba beans (0.207 mg) were significantly heavier than those fed on barley (0.178 mg) and maize (0.189 mg) (*p* < 0.001) (Figure 1A). Female pupae fed on faba beans (0.195 mg) were significantly heavier than those fed on barley (0.163 mg) and maize (0.182 mg) (*p* < 0.001) (Figure 2B). Male pupae were relatively heavier compared to female pupae fed on the same host plant. Male pupae (0.207 mg) were significantly heavier compared to female pupae (0.176 mg) on faba beans (*p* < 0.001) (Figure 1C). Therefore, the variations in pupae weight could be due to variations in nutritional content among the host plants.

### 2.3. Population Parameters of S. frugiperda

The host plants evaluated showed a substantial effect on whether a newly hatched neonate of *S. frugiperda* survived to age *x* and stage *j* (Figure 2). Due to the variations in the developmental rates among *S. frugiperda* individuals, clear overlaps between stages were observed among the host plants. However, a relatively higher survival rate of all developmental stages was observed in maize and wheat than in barley and soya beans. The survival rate of *S. frugiperda* larvae was highest when fed on maize (Figure 2C), with 84.21% of the eggs developing into the adult stage, and the lowest was recorded on barley (65.16%) (Figure 2E). The overall survival duration was longer in adult males than in adult females for *S. frugiperda* fed on all of the host plants (Figure 2A–E). Additionally, on each host plant, female adults appeared 1–2 d earlier than males (Figure 2A–E).

### 2.4. Reproduction Parameters of S. frugiperda

The adult pre-oviposition period (APOP), total pre-oviposition period (TPOP), oviposition period, and fecundity of *S. frugiperda* fed on the five host plants are indicated in Table 3. The APOP of *S. frugiperda* fed on soya beans (4.08 d) was significantly longer than on maize leaves (2.84 d) and barley (3.06 d) (*p* < 0.001). The TPOP of *S. frugiperda* fed on faba beans (37.63 d) and barley (35.26 d) were significantly longer than those fed on other host plants. The oviposition period of *S. frugiperda* fed on maize (6.16 d) and faba beans (6.09 d) was significantly longer than those fed on wheat (4.58 d) and barley (4.97 d) (*p* < 0.001). The fecundity of *S. frugiperda* fed on faba beans (1706.40) and maize (1705.45) was significantly higher than those fed on other host plants (*p* < 0.001). The female ratio of adult *S. frugiperda* was highest for soya beans (53.95%) and lowest for barley (30.49%).

### 2.5. Population Parameters of S. frugiperda

The effects of the five host plants on the population parameters of *S. frugiperda* are indicated in Table 4. The intrinsic rate of increase (*r* = 0.205 d^−1^) and finite rate of increase (λ = 1.228 d^−1^) of *S. frugiperda* fed on maize were significantly higher than those fed on the other host plants but were lowest on faba beans (r = 0.162 d^−1^ and λ = 1.176 d^−1^) and barley (*r* = 0.165 d^−1^ and λ = 1.179 d^−1^) (*p* < 0.01). Moreover, the net reproductive rate (*R_o_*) of *S. frugiperda* fed on maize (695.64) was significantly higher than those fed on wheat (443.43) and barley (397.74) (*p* < 0.01). However, the mean generation time (*T*) of *S. frugiperda* fed on faba beans (39.09 d) was significantly longer than for those fed on the other host plants but was shortest on maize (31.877 d) (*p* < 0.001). Thus, the overall results of population parameters indicated that barley and faba beans were the least preferred host plants for *S. frugiperda* population growth.

### 2.6. Population Survival Rate and Fecundity of S. frugiperda

The impact of the five host plants on *S. frugiperda* age-specific survival rate (*l_x_*), female age-stage specific fecundity (*f_x_*), age-specific fecundity (*m_x_*), and age-specific net maternity value (*l_x_*m_x_*) are displayed in (Figure 3A–E). The *l_x_* curve on the five host plants indicated a decreasing trend as age increased, and the deaths of the last adults on faba bean, soya bean, maize, wheat, and barley were at 57, 52, 47, 48, and 49 d, respectively. The highest *f_x_*, *m_x_*, and *l_x_***m_x_* maximum peaks were attained at 31 d on maize (356.20, 166.98, and 140.61, respectively). However, the lowest *f_x_*, *m_x_*, and *l_x_***m_x_* maximum peaks were attained at 37 d, on barley (152.94, 88.54, and 56.71, respectively). Furthermore, the *f_x_* curve on maize and soya beans had one peak, while there were two or more peaks on the other host plants, indicating that there were significant variations in adult emergence and oviposition period among *S. frugiperda* individuals.

### 2.7. Life Expectancy of S. frugiperda

The age-stage life expectancy (*e_xj_*) represents the length of time that *S. frugiperda* individuals of age *x* and stage *j* are expected to survive after age *x* (Figure 4A–E). The value of *e_xj_* showed a decreasing trend on all host plants, and the average life expectancy values of *S. frugiperda* individuals that were fed on faba beans, soya beans, maize, wheat, and barley were 40.69, 34.72, 34.51, 34.49, and 33.71 d, respectively.

### 2.8. Reproduction Value of S. frugiperda

The age-stage specific reproductive values (*v_xj_*) of *S. frugiperda* indicated the role of an individual at age *x* and stage *j* in upcoming populations (Figure 5A–E). The *v_xj_* value substantially increased when the *S. frugiperda* adults started laying eggs. Increases in *v_xj_* happened at 30–35, 27–33, 24–30, 26–30, and 28–31 d on faba beans, soya beans, maize, wheat, and barley, respectively. The reproductive peaks on maize and wheat occurred slightly earlier, at 30 d, with the highest peak values of 968.69 and 573.65, respectively. The reproductive peak on faba beans occurred later at 35 d, with the highest peak of 997.89.

## 3. Discussion

*S. frugiperda* is a polyphagous and invasive pest that feeds on more than 353 host plants and belongs to 76 families [11,13]. It damages economically important crops such as maize, rice, sorghum, soya beans, and cotton [28,29]. Variations in host plant species affect the survival rate, growth, developmental period, and reproduction potential of phytophagous insects including *S. frugiperda* [20,25,30]. Every plant species has a variety of secondary metabolic and nutritional substances with unique defensive properties, including tolerance, antibiosis, antixenosis, and combinations of the three mechanisms [31,32]. We investigated the performance of *S. frugiperda* fed on maize, wheat, barley, faba beans, and soya beans and performed analysis using the age-stage two-sex life table approach to determine the computability of these plant species for *S. frugiperda* larvae.

Our research revealed that *S. frugiperda* populations completed their life cycle on each of the five host plants, although there were significant differences in the survival rate, developmental period, reproduction, and population growth when fed on the different host plants. According to earlier research from [20,33,34], maize is optimal for *S. frugiperda* survival and development because it has shorter larval, prepupal, and pupal durations, as well as a greater survival rate compared to the other host plants. Compared to the other host plants, maize and faba beans had longer oviposition periods and higher fecundities, which was consistent with earlier research on these two hosts [33]. Additionally, *S. frugiperda* fed on maize exhibited shorter *T* and increased *r*, λ, and *R_o_*, indicating that maize is a suitable food source for *S. frugiperda* survival, development, and fecundity, which is consistent with earlier research from [33,34,35].

The longer larval growth period, higher TPOP, and lower survival rate on barley and faba beans were indicators of poor larval performance compared to the other host plants examined. Due to the long-life cycles and decreasing number of generations, the lengthier larval stages suggest an inappropriate host for insect growth and development [36]. A lower rate of the initial larvae population developed into adults in barley (65.16%) and faba beans (71.03%), suggesting that those plant species may be unsuitable for *S. frugiperda*. Low protein content, challenges with nutrient ingestion and absorption, and physical and chemical characteristics of these plants could all play a role in the lower larval performance [5,34,37]. Moreover, *S. frugiperda* fed on barley and faba beans led to longer *T*, and decreased *r* and λ, indicating that those host plants are unfavorable for *S. frugiperda* survival and development, which is consistent with findings from [33,38]. The prolonged TPOP of *S. frugiperda* on faba beans and barley contributed to the lower *r* and λ values. This is most likely caused by the presence of some harmful chemicals and a lack of essential nutrients in the plants [39]. The lowest *R_o_* and fecundity were recorded on barley and wheat, indicating that the nutrients from barley and wheat were less supportive for *S. frugiperda* reproduction. However, prior research has demonstrated that *S. frugiperda* has a higher *R_o_* and fecundity when fed on wheat than on pulse crops [20,33].

The highest pupa weight, longest duration of the larval stage, and highest adult longevity was recorded when *S. frugiperda* was fed on faba beans. This contradicted earlier findings that showed *S. frugiperda* fed on faba beans resulted in lower pupal weight and shorter adult longevity than maize and wheat [6,20,33]. Larvae raised on greater carbohydrate-content plants develop a heavier pupal weight, and adults with prolonged larval and pupal development are more resistant to desiccation and hunger for longer periods compared to larvae raised on protein-rich plants [40,41]. The increased duration of the larval developmental period when reared on quality host plants can be a compensation mechanism to gain additional pupal weight and complete its life cycle [34]. The pupal weight and fecundity of *S. frugiperda* fed on faba beans were higher compared to the other host plants, indicating a positive relationship between the two parameters. Male pupa weights of *S. frugiperda* fed on the five host plants were greater than female pupa weights. Intriguingly, our findings indicated that female *S. frugiperda* pupae emerged 1–2 d earlier than male pupae, which is in line with other observations [6,33,42]. We hypothesized that this could be to female *S. frugiperda* migrating early in search of food and oviposition sites.

The *s_xj_* curves showed an overlapping tendency due to variations in developmental rates among *S. frugiperda* individuals, which is comparable to prior reports from [6,9,20,22,33,43]. *S. frugiperda* had a higher survival rate on maize (84.21%) than it did on barley (65.16%). *S. frugiperda* individuals of the same age in different stages showed variability in *e_xj_* and *v_xj_* values, which is consistent with observations from [6,20]. The *e_xj_* values showed a declining tendency on all of the host plants; the longest average *e_xj_* values of *S. frugiperda* individuals were on faba beans (40.69 d) and the shortest value was on barley (33.71 d). Maize (968.69) and wheat (573.65) had slightly earlier *v_xj_* peaks at 30 d, whereas the *v_xj_* peaks on faba beans (997.89) had slightly later *v_xj_* peaks at 35 d. The *e_xj_* is examined using the *s_xj_,* assuming that the population receives a constant age-stage distribution. Therefore, it could be useful for predicting the population’s survival in that situation. To create effective control strategies, it is critical to accurately predict future *S. frugiperda* populations.

## 4. Materials and Methods

### 4.1. Insects and Host Plants

Corn strain *S. frugiperda* that had been raised for more than ten generations in a Key Laboratory of Green Prevention and Control of Agricultural Transboundary Pests of Agriculture Environment and Resources Institute, Yunnan Academy of Agricultural Sciences (YAAS), in Yunnan Province, China, was used for the experiment. Insects were maintained at 25 ± 1 °C, 70 ± 5% relative humidity (RH), and 16 h L:8 h D photoperiod. Leaves of five plants species: maize (*Zea mays* L.), wheat (*Triticum aestivum* L.), barley (*Hordeum vulgare*), faba beans (*Visia faba*), and soya beans (*Glycine max*) were used in the population growth and life table study of *S. frugiperda*. Seeds of all plant species were collected from YAAS. The collected seeds of individual plant species were planted separately in plastic pots (10 × 15 cm^2^) filled with a 3:1:1 combination of commercial peat. All plants were maintained in the same climate-controlled room. We used 25 d old wheat and barley, 15 d old maize, 20 d old faba beans, and soya beans seedlings, which were selected based on the growth cycle and planting conditions in the field of these plant species [33]. 

### 4.2. Population Growth and Life Table of S. frugiperda

The population growth and life table study of *S. frugiperda* fed on five host plants was performed using previously outlined methods [20,33,42]. Approximately 180 eggs of *S. frugiperda* laid within 6 h were removed from the growth chamber and placed in a Petri dish (12.0 cm in diameter and 2.0 cm in height) until hatching began. The filter paper was placed at the bottom of the petri dish, and a small drop of water was dropped to maintain the required level of humidity (about 65–75% RH). Every 6 h, the eggs were checked, and the number of hatching larvae was recorded. Single first instar larvae were transferred from the petri dish to a plastic cup (3 × 4 × 3.5 cm^3^) with tiny holes using a soft camel hairbrush. To prevent microbial contamination, the leaves that were given to the larvae were replaced every 24 h. In total, 76, 97, 89, 107, and 105 neonates were used for maize, wheat, barley, faba beans, and soya beans, respectively. Survival and development of the larvae were monitored and recorded daily. Each freshly pupated larva was collected once every 24 h and weighed using an electronic balance. After being sexed, each pupa was placed in a cotton-lined plastic cup. Daily inspections of the pupae were made until adult emergence. The newly emerged adults from the same host plant were paired and put into separate transparent plastic cylindrical boxes (8.5 × 6 cm^2^). For adult nutrition, a cotton ball dipped in a 10% honey-water solution was used. Folded buffer paper was inserted as an oviposition substrate. The cotton ball and buffer paper were changed every day until the adults’ demise. If the male died, a new male from the same mass-reared colony was inserted until the female died. Newly deposited eggs were collected and counted daily until the adults’ demise. All tests were performed in a growth chamber at 25 ± 1 °C, 70 ± 5% RH, and 16 h light: 8 h darkness.

### 4.3. Life Table Data Analysis

The life table data of *S. frugiperda* were estimated using the TWO-SEX-MSChart program [44] according to the age-stage, two-sex life table technique [45,46]. The age-specific survival rate (*s_xj_*) (*x* = age, *j* = stage), which is the likelihood that a newly laid egg survives to age *x* and stage *j*, and fecundity *f_xj_*, estimates the number of eggs deposited by an adult female at *x*. Age-specific fecundity (*m_x_*), age-specific maternity (*l_x_m_x_*), intrinsic rate of increase (*r*), finite rate of increase (λ), net reproductive rate (*R_o_*), and mean generation time (*T*) were analyzed. Age-specific survival rate (*l_x_*): the probability that a newly deposited egg will survive to age *x* was estimated as:(1)$$l_{x}=\overset{m}{\underset{j=1}{\sum }}s_{xj\;}\;$$
where *m* is the number of stages.

Age-specific fecundity (*m_x_*): the number of eggs per individual at age ***x*** was estimated as:(2)$$m_{x}=\overset{m}{\underset{j=1}{\sum }}s_{xj}f_{xj}∕\overset{m}{\underset{j=1}{\sum }}s_{xj}f_{xj}\;$$

The intrinsic rate of increase (*r*) was estimated using the Euler–Lotka equation with age indexed from 0 as follows [47].
(3)$$\overset{\infty }{\underset{x=0}{\sum }}e^{-r\left(x+1\right)/m_{x}=1}$$

The net reproductive rate (*R_o_*), which is defined as the total number of offspring that an individual female adult can have over their lifetime, was estimated as:(4)$$R_{o}=\overset{\infty }{\underset{x=0}{\sum }}l_{x}m_{x}$$

The finite rate (ƛ) is estimated as follows:(5)$$ƛ=e^{r}$$

The mean generation time (*T*) shows how long it takes for a population to grow to *R_o_*-fold of its current size as the time approaches infinity and the population settles to a stable age-stage distribution. Mean generation time was estimated as follows:(6)$$T=\frac{\;\ln R_{o}\;}{r}$$

Age-stage specific life expectancy (*e_xy_*) (i.e., the time that an individual of age *x* and stage *y* is expected to live) was calculated using the method described by [48]:(7)$$e_{xy}=\overset{n}{\underset{i=x}{\sum }}*\;\overset{m}{\underset{j=y}{\sum }}s\prime_{ij}$$
where *s_ij_* is the likelihood that an individual of age *x* and stage *y* will survive to age *i* and stage *j*.

Age-stage-specific reproductive value (*V_xj_*) is the contribution of individuals of age *x* and stage *y* to the future population and was estimated as follows:(8)$$V_{xy}=\frac{e^{-r\left(x+1\right)}}{s_{xy}}\overset{n}{\underset{i=x}{\sum }}e^{-\left(i+1\right)}\overset{m}{\underset{j=y}{\sum }}s\prime_{ij}f_{ij}$$

The means and standard errors were analyzed via the bootstrap method with 100,000 repeats [49]; and the variations among treatments were analyzed using a paired bootstrap test [6,50]. All graphics were analyzed using Graphic Pad Prism 8.0 tool.

## 5. Conclusions

The life cycle of *S. frugiperda* populations was completed on all five host plants, but variations were observed in survival rate, development, reproduction, and population growth among the host plants. The shortest larval and pupal durations, APOP and TPOP, and highest survival rate were recorded for *S. frugiperda* fed on maize. The longest oviposition period and the highest fecundity were recorded on maize and faba beans. In addition to the prolonged oviposition period, *S. frugiperda* had the highest pupa weight and longest adult longevity when fed on faba beans, factors that could contribute to higher fecundity. The lowest survival rate, longest larval period, TPOP and *T*, and decreased *r* and λ were shown for *S. frugiperda* fed on barley and faba beans, indicating that those host plants were unsuitable for *S. frugiperda* survival and development. The lowest *R_o_* and fecundity were shown on barley and wheat compared with the other host plants, indicating that those host plants were less supportive for *S. frugiperda* reproduction. The *s_xj_* curves showed an overlapping trend due to differences in the developmental rates among *S. frugiperda* individuals. The *e_xj_* values showed a decreasing trend on the five host plants, while the longest average *e_xj_* values were on faba beans (40.69 d) and the shortest was on barley (33.71 d). The overall result indicated that *S. frugiperda* had a high preference for maize and a low preference for barley and faba beans, which could help forecast the population’s survival. In conclusion, it is important to precisely forecast the development of *S. frugiperda* populations to establish effective control practices.

## Figures and Tables

**Figure 1 plants-12-00950-f001:**
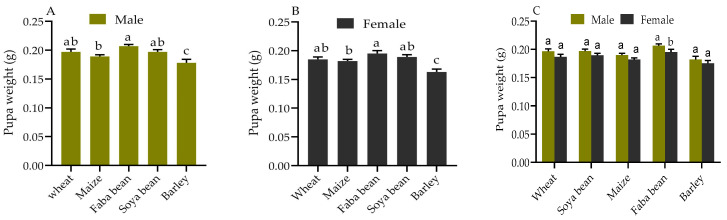
Pupa weight of *S. frugiperda* fed on the five host plants. Different letters (a, b, ab and c) indicate significant differences between pupal weights of males (**A**) and females (**B**) *S. frugiperda* fed on the five host plants (*p* < 0.05, Tukey’s test), and pupal weight of males and females (**C**) *S. frugiperda* fed on the same host plants (*p* < 0.05, Student’s t-test).

**Figure 2 plants-12-00950-f002:**
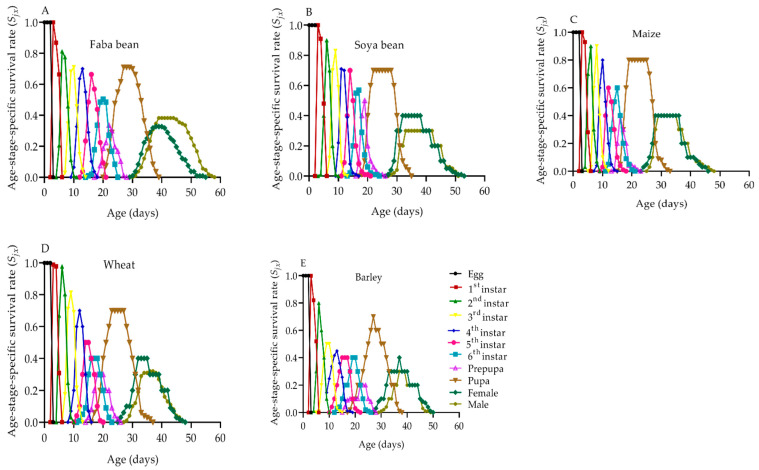
The age-stage specific survival rate of *S. frugiperda* on faba beans (**A**), soya beans (**B**), maize (**C**), wheat (**D**) and barley (**E**). *S_xj_*: the probability that a newly laid egg will survive to age *x* and stage *j*.

**Figure 3 plants-12-00950-f003:**
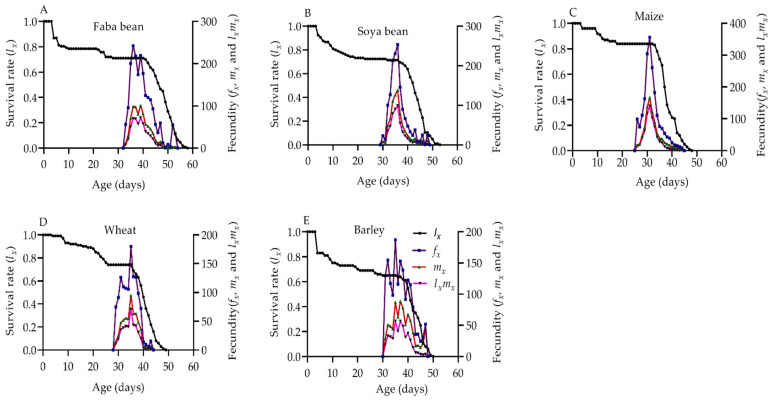
Age-specific survival rate (*l_x_*) and female age-stage specific fecundities (*f_x_*), fecundity (*m_x_*), and net maternity (*l_x_*×(*l_x_*×*m_x_*) of *S. frugiperda* fed on faba beans (**A**), soya beans (**B**), maize (**C**), wheat (**D**), and barley (**E**).

**Figure 4 plants-12-00950-f004:**
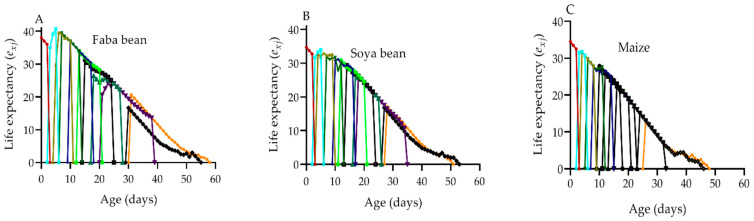
Life expectancy of *S. frugiperda* on faba beans (**A**), soya beans (**B**), maize (**C**), wheat (**D**), and barley (**E**). *e_xj_*: the survival probability of an individual of age *x* and stage *j*.

**Figure 5 plants-12-00950-f005:**
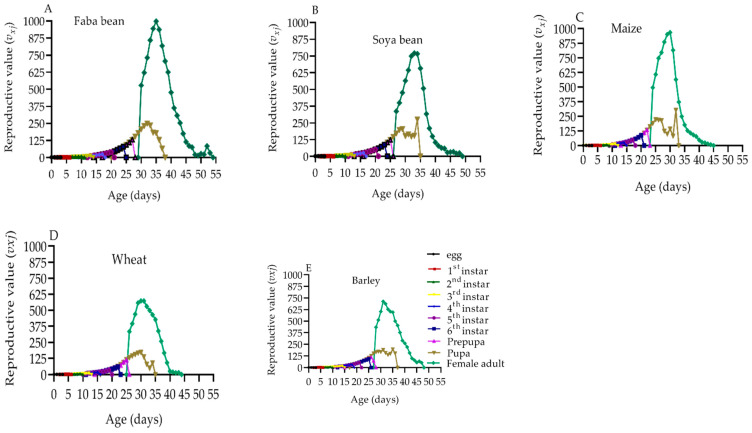
Age-stage specific reproductive value (*v_xj_*) of *S. frugiperda* on faba beans (**A**), soya beans (**B**), maize (**C**), wheat (**D**), and barley (**E**). *v_xj_*: the contribution of an individual of age *x* and stage *j* to future population growth.

**Table 1 plants-12-00950-t001:** Development time and longevity (M ± SE) of *S. frugiperda* fed on five host plants.

Development Stage, d	Host Plants
Maize	Faba Beans	Soya Beans	Wheat	Barley	*p*-Value
Egg	3.00 ± 0.00	3.00 ± 0.00	3.00 ± 0.00	3.31 ± 0.31	3.00 ± 0.0	NS
1st instar	2.26 ± 0.05 c	2.76 ± 0.04 a	2.52 ± 0.05 b	2.32 ± 0.05 c	2.60 ± 0.06 b	<0.0001
2nd instar	2.11 ± 0.04 c	2.87 ± 0.08 a	2.53 ± 0.06 b	2.79 ± 0.05 a	2.79 ± 0.68 a	<0.0001
3rd instar	2.15 ± 0.05 c	3.07 ± 0.08 a	2.81 ± 0.05 b	2.86 ± 0.05 b	2.89 ± 0.77 ab	<0.0001
4th instar	2.25 ± 0.05 d	3.45 ± 0.06 a	2.70 ± 0.07 c	2.83 ± 0.07 c	3.14 ± 0.63 b	<0.0001
5th instar	2.41 ± 0.07 d	3.50 ± 0.07 a	2.65 ± 0.07 c	2.54 ± 0.07 cd	3.17 ± 0.60 b	<0.0001
6th instar	2.54 ± 0.07 b	3.24 ± 0.08 a	2.50 ± 0.06 b	2.63 ± 0.07 b	3.24 ± 0.11 a	<0.0001
1st to 6th instar	13.72 ± 0.18 d	18.94 ± 0.19 a	15.64 ± 0.15 c	15.93 ± 0.21 c	17.86 ± 0.31 b	<0.0001
Pre-pupa	1.40 ± 0.06 c	2.05 ± 0.04 b	1.97 ± 0.04 b	1.97 ± 0.05 b	2.27 ± 0.63 a	<0.0001
Pupa	9.69 ± 0.11 c	10.75 ± 0.12 a	10.32 ± 0.10 b	10.37 ± 0.14 b	9.95 ± 0.89 c	<0.0001
Pre-adult	27.83 ± 0.22 d	34.79 ± 0.26 a	30.92 ± 0.20 c	30.97 ± 0.28 c	33.05 ± 0.35 b	<0.0001
Female adult	11.94 ± 0.42 bc	12.86 ± 0.45 ab	13.51 ± 0.47 a	11.21 ± 0.39 c	11.15 ± 0.46 c	0.0002
Male adult	10.58 ± 0.54 c	16.07 ± 0.42 a	13.66 ± 0.47 b	9.94 ± 0.45 c	10.04 ± 0.63 c	<0.0001
All Adult ( 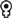 + 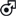 )	11.26 ± 0.35 c	14.59 ± 0.36 a	13.58 ± 0.33 b	10.62 ± 0.30 c	10.59 ± 0.38 c	<0.0001
Total longevity ( 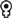 )	39.13 ± 0.59 d	46.69 ± 0.58 a	43.78 ± 0.58 b	41.26 ± 0.53 c	43.35 ± 0.63 b	<0.0001
Total longevity ( 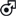 )	39.00 ± 0.67 d	51.68 ± 0.47 a	45.34 ± 0.62 b	41.97 ± 0.51 c	44.29+0.49 b	<0.0001
Total longevity ( 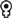 + 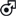 )	39.06 ± 0.44 d	49.38 ± 0.47 a	44.50 ± 0.43 b	41.59 ± 0.37 c	43.74 ± 0.42 b	<0.0001

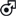
 = Male, 
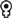
 = Female. The larval development stages, pupal, pre adult and adult durations, and total longevity were analyzed using age-stage, two-sex life table. Mean values M ± SE in the same row followed by different letters were significantly different (*p* < 0.05) (paired bootstrap test).

**Table 2 plants-12-00950-t002:** Survival rate at each stage of *S. frugiperda* fed on maize, faba beans, soya beans, wheat, and barley before adult emergence.

Survival Rate, %	Host Plants				
	Maize	Faba Beans	Soya Beans	Wheat	Barley
Egg	100.00	100.00	100.00	100.00	100.00
1st instar	96.05	86.91	90.47	98.97	83.14
2nd instar	96.05	79.43	86.66	98.97	78.65
3rd instar	94.74	78.50	80.00	92.78	74.15
4th instar	88.16	78.50	77.14	90.72	73.03
5th instar	86.86	78.50	75.24	89.70	73.03
6th instar	85.53	78.50	74.29	88.66	70.78
Prepupa	85.53	78.50	73.33	84.53	67.42
Pupa	84.21	71.03	72.30	73.20	65.16
Adult	84.21	71.03	72.30	73.20	65.16

The data in the table are percentage survival rate of S. frugiperda at each developmental stages.

**Table 3 plants-12-00950-t003:** Adult pre-oviposition period (APOP), total pre-oviposition period (TPOP), oviposition d, fecundity (M ± SE), and female ratio of *S. frugiperda* fed on the five host plants.

Host Plants	Biological Parameters				
	APOP (d)	TPOP (d)	Oviposition (d)	Fecundity	Female Ratio
Maize	2.84 ± 0.22 c	30.02 ± 03 d	6.16 ± 0.34 a	1705.45 ± 125.84 a	48.44%
Faba beans	3.80 ± 0.37 ab	37.63 ± 0.50 a	6.09 ± 0.29 a	1706.40 ± 123.19 a	46.05%
Soya beans	4.08 ± 0.24 a	34.30 ± 0.34 bc	5.61 ± 0.34 ab	1315.59 ± 95.78 b	53.95%
Wheat	3.57 ± 0.20 abc	33.54 ± 2.55 c	4.58 ± 0.25 c	1015.29 ± 89.98 c	36.54%
Barley	3.06 ± 0.17 bc	35.26 ± 2.91 b	4.97 ± 0.28 bc	1160.74 ± 59.05 bc	30.49%
*p*-value	<0.0102	<0.0001	<0.0007	<0.0001	

The reproduction parameters (APOP, TPOP, oviposition period and fecundity) were analyzed using age-stage, two-sex life table. Mean values M ± SE in the same column followed by different letters were significantly different (*p* < 0.05) (paired bootstrap test).

**Table 4 plants-12-00950-t004:** Net reproductive rate (*R_o_*), intrinsic rate of increase (*r* (d^−1^)), finite rate of increase (λ (d^−1^)), and mean generation time (*T* (d)) of *S. frugiperda* fed on five host plants.

Host Plants	Population Parameters			
	*R_o_*	*r* (d^−1^)	λ (d^−1^)	*T* (d)
Maize	695.64 ± 108.34 a	0.205 ± 0.0065 a	1.228 ± 0.0074 a	31.877 ± 0.44 c
Faba beans	558.17 ± 86.85 ab	0.162 ± 0.0049 b	1.176 ± 0.0057 b	39.079 ± 0.45 a
Soya beans	513.70 ± 72.83 ab	0.173 ± 0.0046 b	1.188 ± 0.0054 b	36.167 ± 0.34 bc
Wheat	443.43 ± 63.66 b	0.172 ± 0.0054 b	1.187 ± 0.0063	34.92 ± 0.47 c
Barley	397.74 ± 61.11 b	0.165 ± 0.0047 b	1.179 ± 0.0053 b	36.93 ± 0.52 b
*p*-value	<0.0001	<0.0001	<0.0001	<0.0001

Population parameters *R_o_*, *r* (d^−1^), λ (d^−1^) and *T* (d) were analyzed using age-stage, two-sex life table. Mean values M ± SE in the same column followed by different letters were significantly different (*p* < 0.05) (paired bootstrap test).

## Data Availability

Data can be provided upon request from the lead author.

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
