# Peer review of "Population Growth of Fall Armyworm, Spodoptera frugiperda Fed on Cereal and Pulse Host Plants Cultivated in Yunnan Province, China"

_plants, 2023, doi:10.3390/plants12040950_

Round 1

Reviewer 1 Report

The Manuscript is interested but need a lots of work to improve it. The main issue is that the manuscript consists of 35% of the Similarity index, i would strongly recommend to rephrase all the MS especially the introduction, methods, and discussion section. The MS should have less than 19% of the similarity index.

1. the introduction is outdated and do not fitful the need of the title of manuscript, it can be improved and must have recent knowledge importance and information. 

2. Rewrite the last Paragraph of the introduction and explain the rational's and goals , and how you bridged the goals. also mention you novelty , what is novel, as number of previous paper has been reported these topic. 

Material method Section needs to revised extensively with updated citations and correctness. 

Numbers of Citation is missing

Author Response

Dear Editor and Reviewers,

We much appreciate your favorite consideration and reviewers’ insight comments on our manuscript of “Population Growth of fall armyworm, Spodoptera frugiperda fed on Cereal and Pulse Host Plants Cultivated in Yunnan Province, China”. We have studied each comment carefully and revised the paper according to the editors and reviewers. We hope this revision can make our paper acceptable. The main corrections in the paper and responses to the reviewers’ comments are as following:

Responses to Reviewer

  • English language edited by native speaker.
  • Based on the general comments and suggestions; we tried to rephrase almost all parts of the paper (mainly introduction, methods and discussion).
  1. The introduction is outdated and do not fitful the need of the title of manuscript, it can be improved and must have recent knowledge importance and information?

Response: Based on the suggestion, we improved introduction, and included same recent findings that can fit to the topic.

  1. Rewrite the last Paragraph of the introduction and explain the rationale’s and goals, and how you bridged the goals. also mention your novelty, what is novel, as number of previous papers has been reported these topics.

 Response: we revised the last paragraph of the introductory part based on the revisers suggested (page 3, Line 106-121).

  1. Material method Section needs to revised extensively with updated citations and correctness.

Response:  Based on the suggestion Materials and methods revised and included same updated references (Page 11- 14, Line 353-467).

  1. Numbers of Citation is missing.

Response:  Number of citations checked based on the comments given.

Reviewer 2 Report

The manuscript entitled  „Population Growth of fall armyworm, Spodoptera frugiperda fed on Cereal and Pulse Host Plants Cultivated in Yunnan Province, China”, written by Gebretsadik and co-workers contains important information regarding development on five agriculturally important crop plants. The Authors analyzed the following indicators for polyphagous insect pest S. frugiperda as: larvae stage, larva-adult period, adult pre-oviposition period (APOP), total pre-oviposition period (TPOP), and generation time (T), and survival rate, intrinsic (r) and finite (λ) rate and net reproductive rate (R0). The article should be published because the results provide new information on the pest, which damaged many economic crops and has the ability to spread rapidly.

My comments are below:

Results showing significant differences in table 1,  they should be more exposed/highlighted.

In table 1, "Development stage" should be used instead of "Duration, d"

Author Response

Dear Editor and Reviewers,

We much appreciate your favorite consideration and reviewers’ insight comments on our manuscript of “Population Growth of fall armyworm, Spodoptera frugiperda fed on Cereal and Pulse Host Plants Cultivated in Yunnan Province, China”. We have studied each comment carefully and revised the paper according to the editors and reviewers. We hope this revision can make our paper acceptable. The main corrections in the paper and responses to the reviewers’ comments are as following:

Responses to Reviewer

  1. Results showing significant differences in table 1, they should be more exposed/highlighted.

Response: The significance levels revised as "Asterisk indicates the significant di?erence (*, P?0.05; **, P?0.01, ***, P?0.001)".

  1. In table 1, "development stage" should be used instead of "duration, d"?

Response: Based on the suggestion, "duration, d" replaced by "development stage".

Note: English language edited by native speaker.

Reviewer 3 Report

The results showed that Spodoptera frugiperda had highly preference to maize and lowly preference to barley and faba bean.  It is important to precisely forecast the developmental approach of the S. frugiperda populations. Could be interesting to find some predators and parasitoids for biological control with S. frugiperda.

Author Response

Dear Editor and Reviewers,

We much appreciate your favorite consideration and reviewers’ insight comments on our manuscript of “Population Growth of fall armyworm, Spodoptera frugiperda fed on Cereal and Pulse Host Plants Cultivated in Yunnan Province, China”. We have studied each comment carefully and revised the paper according to the editors and reviewers. We hope this revision can make our paper acceptable. The main corrections in the paper and responses to the reviewers’ comments are as following:

Responses to Reviewer

  1. The results showed that Spodoptera frugiperda had highly preference to maize and lowly preference to barley and faba bean. It is important to precisely forecast the developmental approach of the frugiperda populations. Could be interesting to find some predators and parasitoids for biological control with S. frugiperda

Response: Based on the suggestions and comments, we included the “Could be interesting to find some predators and parasitoids for biological control with S. frugiperda” as recommendation in our future work “(page 14, Line 504-505).

Note: English language edited by native speaker.

Round 2

Reviewer 1 Report

Minor English check is required